



# Characterization of secondary organic aerosol from heated-
# cooking oil emissions: evolution in composition and volatility
Manpreet Takhar[1], Yunchun Li[2], Arthur W. H. Chan[1]
[1]Department of Chemical Engineering and Applied Chemistry, University of Toronto, Toronto, M5S 3E5, Canada
[2]College of Science, Sichuan Agricultural University, Ya'an, 625014, China
*Correspondence to*: Arthur W. H. Chan (arthurwh.chan@utoronto.ca)
**Abstract.** Cooking emissions account for a major fraction of urban organic aerosol. It is therefore important to
understand the atmospheric evolution in the physical and chemical properties of organic compounds emitted from
cooking activities. In this work, we investigate the formation of secondary organic aerosol (SOA) from oxidation of
gas-phase organic compounds from heated cooking oil. The chemical composition of cooking SOA is analyzed using
thermal desorption-gas chromatography-mass spectrometry (TD-GC/MS). While the particle-phase composition of
SOA is a highly complex mixture, we adopt a new method to achieve molecular speciation of the SOA. All the GC
elutable material is classified by the constituent functional groups, allowing us to provide a molecular description of
its chemical evolution upon oxidative aging. Our results demonstrate an increase in average oxidation state (from -0.6
to -0.24), and decrease in average carbon number (from 5.2 to 4.9) with increasing photochemical aging of cooking
oil, suggesting that fragmentation reactions are key processes in the oxidative aging of cooking emissions within 2
days equivalent of ambient oxidant exposure. Moreover, we estimate that aldehyde precursors from cooking emissions
account for a majority of the SOA formation and oxidation products. Overall, our results provide insights into the
atmospheric evolution of cooking SOA, a majority of which is derived from gas-phase oxidation of aldehydes.
## 1   Introduction
Organic aerosol (OA) has important impacts on air quality, climate and human health (Hallquist et al., 2009). OA is
often composed of thousands of organic compounds formed from a variety of sources. In urban areas, particulate
emissions from food cooking account for a significant fraction of OA (Allan et al., 2010; Crippa et al., 2013; Florou
et al., 2017; Kostenidou et al., 2015; Lee et al., 2015; Mohr et al., 2012; Sun et al., 2011). Furthermore, volatile organic
compounds (VOCs) are also emitted, and they can undergo oxidation and form secondary organic aerosol (SOA).
Recent studies have reported the formation of SOA from meat charbroiling (Kaltsonoudis et al., 2017a) and heated
cooking oils (Liu et al., 2017b, 2017c, 2018). Therefore, food cooking activities have substantial impacts on air quality
in and downwind of urban areas.
The emission of VOCs from cooking is highly variable and depends on a number of factors such as cooking style,
food, ingredients, and temperature (Fullana et al., 2004a, 2004b; Klein et al., 2016a, 2016b; Liu et al., 2017c; Schauer
et al., 1999, 2002). Of the different classes of VOCs characterized in these studies, aldehydes have been shown to be



the major group of VOCs emitted from cooking oils. These VOCs are chemically produced upon heating via peroxyl
radical reactions of the fatty acids (Choe and Min, 2007; Gardner, 1989). Klein et al. (2016a) investigated the
composition of nonmethane organic gas (NMOG) emissions from boiling, charbroiling, shallow and deep frying of
various vegetables, meats, and cooking oils heated under different temperature conditions. The authors reported that
emissions from shallow frying, deep frying and charbroiling are dominated by aldehydes, and the relative amounts
depend on the type of oil used during cooking (Klein et al., 2016a). C7 aldehydes are the major species in emissions
from canola oil, whereas C9 aldehydes are dominant from olive oil (Klein et al., 2016a). These differences in emission
patterns of oils vary with composition of triglycerides present in the oil (Choe and Min, 2006). Katragadda et al.
(2010) demonstrated up to an order of magnitude increase in emissions upon reaching the smoke point of cooking
oils. In addition to emissions from cooking oil, the addition of condiments (herbs and peppers) to cooking leads to
significant emissions of mono-, sesqui- and diterpenes in the gas phase (Klein et al., 2016b). Liu et al. (2017a) showed
an order of magnitude increase in the emissions of VOCs when stir-frying with spices. Therefore, factors like cooking
style, food, cooking temperature, and ingredients play a significant role in the chemical profile of cooking emissions
(Fullana et al., 2004a, 2004b; Klein et al., 2016a, 2016b; Liu et al., 2017a, 2017c).
The VOCs emitted from cooking have been shown to produce significant amount of SOA rapidly in recent flow tube
(Liu et al., 2017b) and smog chamber studies (Kaltsonoudis et al., 2017a; Liu et al., 2017c, 2018). Kaltsonoudis et al.
(2017a) and Liu et al. (2017b, 2018) showed an increase in O/C ratio upon a few hours of atmospheric aging suggesting
lightly oxidized cooking SOA. Furthermore, Liu et al. (2017b) showed significant production of SOA with increasing
OH exposure for different cooking oils. Thus far studies have only focused on formation potential of SOA from
cooking emissions. Despite high emission rates of VOCs from cooking, the understanding of SOA composition from
these emissions remains limited.
Source apportionment using aerosol mass spectrometry (AMS) data in urban areas has often revealed a Cooking
Organic Aerosol (COA) factor, but it is unclear how this factor is related to cooking emissions. Many studies reported
that the mass spectra associated with this factor resemble that of hydrocarbon-like organic aerosol (HOA) factor from
other non-cooking sources (Dall'Osto et al., 2015; Hayes et al., 2013; Huang et al., 2010; Mohr et al., 2009, 2012). In
addition, it is often unclear whether ambient COA represents primary or secondary organic aerosol from cooking
emissions (Dall'Osto et al., 2015; Florou et al., 2017; Kaltsonoudis et al., 2017b; Kostenidou et al., 2015). Laboratory
studies (Liu et al., 2017b, 2018) showed that the mass spectra for primary cooking organic aerosol exhibited strong
correlation with ambient COA factor (Lee et al., 2015), but the cooking SOA mass spectra showed some similarities
to ambient semi-volatile oxygenated OA (SV-OOA) factor. These measurements highlight the challenges in assigning
COA factor without understanding the changes in chemical composition occurring during oxidation of cooking
emissions.
In general, there is a need to better understand the molecular composition contributing to aged COA. In this study, we
investigate detailed chemical composition of cooking SOA at the molecular level. The objectives of this study are to:
(i) understand the detailed chemical speciation of cooking SOA using TD-GC/MS, (ii) describe chemical evolution in
SOA upon atmospheric aging, and (iii) attribute formation of SOA to different VOCs emitted from food cooking



emissions. In this work we use heated cooking oil as a model for food cooking emissions. We show that the majority
of the SOA is derived from oxidation of aldehydes, and the oxidation mechanisms are dominated by fragmentation
reactions. Overall, our results provide useful insights into the evolution of cooking SOA, which may be incorporated
into chemical transport models for better predicting OA formation from cooking emissions in the atmosphere.

## 2    Experimental methods
### 2.1    Flow tube experiments
The experimental setup is shown in Fig. 1, and experimental conditions are listed in Table S1. For each experiment,
30-40 mL of canola oil was heated at 250 °C on an electric heating plate in a Pyrex bottle resulting in an average
cooking oil temperature of 180 °C, as measured by a thermocouple in direct contact with the heated oil. Purified air
flowed over the headspace of the heated oil at a rate of 0.2 L min$^{-1}$ and then diluted by a factor of 50. 0.2 L min$^{-1}$ of
the total diluted flow was passed through a Teflon filter to remove particles, and the oil vapors were introduced into a
custom-built 10 L quartz flow tube reactor. A separate flow of oxygen (99.6%) was irradiated in a UV ozone generator
(UVP 97006601) to produce ozone and was also introduced into the flow tube reactor. In parallel, purified air was
flowed through a water bubbler into the reactor to provide water vapor. The combined flow rate through the flow tube
was set at 3 L min$^{-1}$, resulting in an average residence time of approximately 200 s.
In the flow tube, hydroxyl radicals were produced through the photolysis of ozone irradiated by a UV lamp ($\lambda = 254$
nm) in the presence of water vapor. The integrated OH exposure was measured indirectly from the loss of cyclopentane
which was monitored by a gas chromatography-flame ionization detector (GC-FID, model 8610C, SRI Instruments
Inc.) equipped with a Tenax TA trap sampling downstream of the flow tube at a rate of 0.15 L min$^{-1}$. In this study, the
experiments were conducted at different OH exposures ranging from $5.77 \times 10^{10}$ to $2.2 \times 10^{11}$ molecules cm$^{-3}$ s. OH
exposure in this range is equivalent to ~11 to 41 h of atmospheric oxidation, respectively, assuming a 24-h average
atmospheric OH concentration of $1.5 \times 10^{6}$ molecules cm$^{-3}$ (Mao et al., 2009).
Downstream of the flow tube, pre-baked quartz fiber filter and Tenax tube samples were collected for offline chemical
analysis. The changes in the particle size distribution and volume concentration were monitored using a scanning
mobility particle sizer (SMPS) with a differential mobility analyzer (TSI 3081), and a condensation particle counter
(TSI 3781). A constant density of 1.4 g cm$^{-3}$ was assumed to convert particle volume concentration into mass
concentration (Chan et al., 2010). Relative humidity and temperature were monitored by an Omega HX94C RH/T
transmitter and were maintained at 65-70%, and 19-20 °C, respectively for all experiments. A fast stepping/scanning
thermodenuder (TD, Aerodyne Inc. Billerica, USA) was also placed downstream of the flow tube to measure SOA
evaporation rates. Details about TD operating conditions and analysis can be found in Takhar et al. (2019). The TD
was only operated during one experiment in which the OH exposure was $9.23 \times 10^{10}$ molecules cm$^{-3}$ s. The SOA was
systematically heated in a TD from 25 °C to 175 °C, and changes in particle volume concentrations and corresponding
mass fraction remaining (MFR) were measured using a SMPS. The SOA size distribution during TD operation and



volatility distribution are shown in Fig. S1 and S2, respectively. A kinetic mass transfer model developed by Riipinen
et al. (2010) was used to interpret the TD data. The inputs to the model are volatility distribution of OA, enthalpy of
vaporization, mass accommodation coefficients. Compound groups are translated into volatility distributions by
binning components according to their saturation concentrations (Donahue et al., 2006). Parameterization for enthalpy
of vaporization was similar to that of Takhar et al. (2019). We assume a surface tension of 0.05 N m$^{-1}$, gas-phase
diffusion coefficients of $5\times10^{-6}$ m$^2$ s$^{-1}$ for all simulations similar to that reported in Riipinen et al. (2010).

## 110    2.2    Chemical characterization of SOA

Tenax tube and quartz filter samples were analyzed separately by thermal desorption gas chromatography mass
spectrometry (TD-GC/MS) for detailed chemical speciation of gas- and particle-phase organic compounds. The
analyses were performed using a thermal desorption system (TDS 3, Gerstel) combined with a gas chromatography
(7890B, Agilent)-mass spectrometer (5977A, Agilent). For gas-phase analysis, concentrations of aldehydes (C7 to
C10 $n$-alkanals, alkenals and alkadienals) collected on Tenax tube samples before photooxidation (downstream of the
flow tube, with lights off) were quantified. For particle-phase analysis, thermal desorption of quartz filters was
performed with $in situ$ derivatization using $N$-trimethylsilyl-$N$-methyl trifluoroacetamide (MSTFA). A known amount
of deuterated 3-hydroxy-1,5-pentanedioic-2,2,3,4,4-d$_5$ acid, and $n$-pentadecane-d$_{32}$ (CDN isotopes) was injected,
respectively, onto quartz filter punches, and Tenax tubes as internal standards before the samples were desorbed in
the TDS. All GC/MS analysis was performed using a non-polar DB5 column (Rxi-5Sil MS, Restek). Details of the
operating parameters (GC column, GC and TDS temperature ramps, MS parameters) can be found in Sect. 1 of SI.
With $in situ$ derivatization, polar organic compounds react rapidly with MSTFA at elevated temperatures during
thermal desorption, and functional groups with acidic hydrogen atoms (such as –OH) are replaced by a less polar
trimethylsilyl (TMS, [-OSi(CH$_3$)$_3$]) group. This reduction in polarity allows the derivatized analyte to elute from a
non-polar column and analyzed by subsequent electron impact (EI) at 70 eV. Derivatized compounds produce a
signature fragment ion at mass-to-charge ($m/z$) 73 (-Si(CH$_3$)$_3^+$) arising from the scission of O-Si bond in R-O-
[Si(CH$_3$)$_3$]. In other words, all derivatized compounds produce ions with $m/z$ 73 during analysis. Therefore, the total
signal at $m/z$ 73 can be taken as the total concentration of organic compounds with at least one hydroxyl group
(including both –OH and –C(O)OH) present in cooking SOA, much like how $m/z$ 57 represents total concentration of
aliphatic compounds in hydrocarbon mixtures (Zhao et al., 2014, 2015). It should be noted that organic peroxides (R–
OOH) were also found to be derivatized, but the major reaction product formed is R-O-[Si(CH$_3$)$_3$] (which is also
formed from R–OH derivatization) as shown in Fig. S3. Here we assume alcohols and acids are the major components,
but will explore the potential role of ROOH on the overall chemical composition in Sect. 3.1.
As shown in Fig. 2, many compounds in cooking SOA contain at least one –OH group and the chromatogram of $m/z$
73 is typical of that for a highly complex mixture or unresolved complex mixture (UCM). Using traditional analytical
techniques like GC/MS it is difficult to deconvolute the UCM. However, knowledge about mass spectral





fragmentation of TMS derivatives can be used to understand the compounds contributing to the UCM. Table S2 shows
a list of compounds containing multiple functional groups e.g. -COOH, -OH resulting in different combinations of
compound classes like dicarboxylic acids, hydroxy acids, hydroxy dicarboxylic acids, and dihydroxy dicarboxylic
acids with different carbon numbers. As mentioned earlier, we acknowledge the potential contribution from ROOH,
but will first assume the functional groups shown in Table S2 here, and consider ROOH in more detail in a later
section. The compound groups shown in Table S2 are expected to be formed from oxidation of aldehydes and be
derivatized by MSTFA. The TMS derivatives of these compounds share common ion fragments in their EI mass
spectra: $m/z$ 73 $[Si(CH_3)_3]^{+}$, 75, 147 $[(CH_3)_2Si=O(CH_3)_3]^{+}$, M-15 $[M-CH_3]^{+}$ (Jaoui et al., 2004, 2005; Yu et al., 1998).
Most importantly, all TMS derivatives exhibit quantifiable peaks at $m/z$ 73 (ubiquitous ion for all derivatives) and M-
15 (ion specific to each compound group, hereby referred to as the pseudo-parent ion). We also obtained the
characteristic ratio of these two ions for each compound group ($f_{M-15/73}$) from NIST mass spectral libraries and from
analyzing authentic standards. To verify the validity of this method, we calculate the total $m/z$ 73 ion signal that is
attributable to these compound groups by taking the chromatograms of the pseudo-parent ion for each compound
group, dividing by its characteristic ratio $f_{M-15/73}$ and then summing across all compound groups as shown in Eq.
151 (1).

$$S_{73,t}^{sum} = \sum_i \frac{S_{M-15,i,t}}{f_{M-15/73,i}} \qquad (1)$$
where $S_{73,t}^{sum}$ is the $m/z$ 73 ion signal at retention time $t$ that is attributable to all compound groups listed in Table S2,
$S_{M-15,i,t}$ is the signal of the pseudo-parent ion for compound group $i$ at retention time $t$, $f_{M-15/73,i}$ is the characteristic
ratio of pseudo-parent ion to $m/z$ 73. This approach is similar to that described in Isaacman-VanWertz et al. (2020).
As shown in Fig. 2, $S_{73,t}^{sum}$ shows excellent agreement with the measured $m/z$ 73 ion signal, suggesting that the $m/z$ 73
signal, which is representative of all TMS derivatives, is almost entirely comprised of contributions from the
compound groups listed in Table S2. This agreement between our bottom-up approach and measured signal provides
confidence that our method is able to provide information about the chemical composition of highly complex mixture.
With the signals from all the pseudo parent ions for all compound groups, the total mass of each compound group was
then calculated using Eq. (2).
$$M_i = \frac{TA_i}{RF_i} \times \frac{1}{f_{M-15/73,i}} \qquad (2)$$
where, $M_i$ is the mass of compound group $i$, $TA_i$ is the total integrated signal of pseudo-parent ion for compound
group $i$ (normalized by the signal of deuterated internal standard), $RF$ is the response factor (calculated from
calibration curves of fatty acids and dicarboxylic acids authentic standards) of compound group $i$, and $f_{M-15/73,i}$ is
the characteristic ratio of pseudo-parent ion to $m/z$ 73 for compound group $i$. A more detailed, step-by-step description
of the procedure can be found in the SI in Sect. 2, and illustrated in Fig. S4 with corresponding uncertainties in the
fitting procedure shown in Fig. S5.






## 3    Results and discussion

### 3.1    Chemical evolution of SOA

As described in Sect. 2.2, components in cooking SOA were classified by functional groups and carbon number. To
describe the overall changes in SOA composition with increasing OH exposure, we use the average carbon oxidation
state ($\overline{OSc}$) as a metric for the evolving composition of a complex mixture undergoing oxidation (Kroll et al., 2011).
Both $\overline{OSc}$ and number of carbon atoms (nc) for each compound group are calculated from the GC-derived chemical
composition. The total mole fraction of C, H and O was calculated for each sample which was then used to calculate
the bulk $\overline{OSc}$ using the Eq. $2 \times O{:}C - H{:}C$ (Kroll et al., 2011). The evolution in this framework for canola oil SOA
is shown in Fig. 3. The bulk $\overline{OSc}$ was observed to increase from -0.6 to -0.24 when OH exposure increased from 5.77
to $22.0 \times 10^{10}$ molecules cm$^{-3}$ s for canola oil SOA. For comparison, Liu et al. (2017b) showed an initial decrease in
$\overline{OSc}$ and O:C, but gradually stabilized at OH exposure greater than $9 \times 10^{10}$ molecules cm$^{-3}$ s. For the $\overline{OSc}$ range reported
here, the $\overline{OSc}$ of cooking SOA falls in the range of SV-OOA as determined from factor analysis of AMS data
(Canagaratna et al., 2015). This degree of oxygenation is greater than that of the COA factor measured by AMS, which
is reported to be around -1.37 (Canagaratna et al., 2015). This difference suggests that the COA factor resolved using
PMF analysis is likely of primary origin and does not represent SOA formed from atmospheric oxidation of cooking
emissions. Furthermore, previous GC/MS analysis showed for POA from cooking oils, an $\overline{OSc}$ of -1.66 (canola oil)
and -1.7 (beef tallow, olive oil) was calculated (Takhar et al., 2019). These observations again suggest that COA factor
measured by AMS is derived of primary cooking emissions.
In addition to carbon oxidation state, knowledge about molecular composition provides further insights into the
oxidation mechanisms. Canola oil SOA at an OH exposure of $5.77 \times 10^{10}$ molecules cm$^{-3}$ s is comprised of long chain
hydroxy acids, whereas at higher OH exposure more short-chain dicarboxylic acids and hydroxy dicarboxylic acids
are produced. As a result, oxidation simultaneously leads to higher $\overline{OSc}$ and lower carbon number on average. Based
on the compounds observable by our technique, this trend suggests that fragmentation reactions are key processes in
the oxidative evolution of cooking emissions. These findings suggest an early onset of fragmentation reactions upon
atmospheric aging of cooking emissions contrary to other SOA systems, such as alkanes and isoprene (Lambe et al.,
2012, 2015), in which fragmentation reactions dominate at later OH exposures ($>5 \times 10^{11}$ molecules cm$^{-3}$ s). Therefore,
predicting OA concentrations from cooking emissions would require earlier fragmentation of SOA in climate and air
quality models.
The compounds observed here can also be compared to previously measured bulk composition using elemental ratios,
such as those presented in a Van Krevelen (VK) diagram (Heald et al., 2010). As shown in Fig. 4, the O:C ratio in our
study ranged between 0.64 and 0.79 when OH exposure increased from $5.77 \times 10^{10}$ to $22.0 \times 10^{10}$ molecules cm$^{-3}$ s.
These ratios are within a factor of 2 than previously reported AMS measurements of cooking oil SOA (Kaltsonoudis
et al., 2017a; Liu et al., 2017b). Furthermore, the H:C versus O:C trend is linear with a slope of -0.19, which lies





between the slope of 0 measured for low-NOx oxidation reported by Liu et al. (2017b) and -0.4 for high-NOx conditions
(Liu et al., 2018). Therefore, based on elemental ratios, the evolution in SOA composition measured in this study is
comparable to that in bulk average properties estimated by AMS. Furthermore, we use 2D-VBS framework developed
by Donahue et al. (2012) to investigate OA chemistry, and understand the evolution of cooking SOA through changes
in the volatility of SOA system. The vapor pressures of the identified compounds are calculated using group
contribution method (Pankow and Asher, 2008) where experimentally determined vapor pressures were unavailable,
and reported in Table S2. The observed compounds in SOA have a broad range of volatilities, since they were formed
from oxidation of a complex ensemble of VOC precursors. As shown in Fig. S6, there is minor decrease in overall
volatility of the mixture (change lies within one decade in C*) irrespective of the presence of peroxides, while $\overline{OSc}$ is
increasing with oxidation. This increase in oxidation state is coincident with increasing fragmentation upon oxidation,
and, as a result, the overall change in the bulk volatility of canola oil SOA is relatively small.
As mentioned earlier in Sect. 2.2, there is a potential to misclassify ROOH as ROH using our current GC/MS method.
Here we further examine the chemical composition by assuming that each -O-[Si(CH$_3$)$_3$] group observed originates
from an -OOH group in the SOA, and to support this argument we show that derivatization of cumene hydroperoxide
(Sigma Aldrich Co.) is observed as TMS of hydroxy-cumene in our system as shown in Fig. S3. It should be noted
that replacing –OH with –OOH results in a higher estimate of O:C (and $\overline{OSc}$) but does not change H:C or carbon #.
Furthermore, since pseudo molecular ion fraction ($f_{M-15/73}$) for organic peroxides (needed for quantification) is
unknown, we assume that it is similar to those presented in Table S2. As shown in Fig. S7, if all observed –OH groups
are –OOH groups, the VK-slope is -0.15 which is similar to -0.19 calculated based on the no-peroxide assumption.
Similarly, Fig. S6 shows that this uncertainty in hydroxyl group identification has negligible effect on estimation of
vapor pressure or volatility in the 2D-VBS framework. Therefore, this potential misclassification of peroxide groups
may lead to an underestimation in O:C and $\overline{OSc}$, but is not expected to affect estimates of volatility and our general
conclusions about the importance of fragmentation reactions. In the future, analytical techniques such as extractive
electrospray ionization time-of-flight mass spectrometry (Lopez-Hilfiker et al., 2019) may be useful to better
understand the composition of peroxides from cooking SOA. While the misclassification of peroxides may have little
impact on the bulk properties such as average O:C ratios, there may be important implications on understanding the
reactivity of the SOA.

**3.2    Evaporation rates of SOA**
The volatility of the SOA is also probed by measuring the evaporation rates in a heated thermodenuder and compared
to the rates expected from the measured composition. In order to derive the evaporation rates from the measured
chemical composition of cooking SOA, we use the kinetic mass transfer model developed by Riipinen et al. (2010).
Among the inputs into the model, the mass accommodation coefficient is a critical but uncertain parameter that
accounts for the mass transfer limitations in the system.



Figure 5 shows both measured and modeled mass thermograms for canola oil SOA. We observe that for canola oil
SOA, mass accommodation coefficient of 0.03 is needed to predict the experimentally determined mass thermograms.
An accommodation coefficient of <1 suggests that mass transfer limitations in the system likely occurring in the
condensed-phase. Formation of multifunctional organic compounds such as those observed in this study is likely
responsible for an increase in viscosity through increasing hydrogen bonding and other polar interactions (Rothfuss
and Petters, 2016). It should be noted that Takhar et al. (2019) reported similar magnitudes of mass accommodation
coefficients for heterogeneous oxidation of cooking oil particles. Due to similarity in the type of functional groups
present in both aging pathways, we believe the decrease in mass accommodation coefficients for both systems undergo
similar changes in phase and/or viscosity.
These measurements of evaporation rates are consistent with the volatilities expected from our measured composition
of SOA containing small oxygenated compounds. Although mass accommodation coefficients are highly uncertain,
the mass accommodation coefficients for other SOA systems have been measured to be even lower on the order of $10^{-4}$
$^{4}$ (Cappa and Wilson, 2011), which would require the volatilities to be even higher to explain the measured evaporation
rates. Therefore, the TD measurements support the conclusion that smaller oxygenated compounds are produced from
oxidation of cooking oil vapors, and that fragmentation reactions are dominant. Furthermore, these measurements
provide useful inputs into chemical transport models for predicting SOA formation and gas-particle partitioning. Our
previous work (Takhar et al., 2019) showed that even at $\alpha = 10^{-2}$, gas-particle partitioning timescales are short (within
hours) and the assumption of equilibrium partitioning still holds for regional scale SOA formation. Further work is
needed to directly measure the viscosity of cooking SOA, and corresponding mixing timescales to better constrain the
physicochemical properties of cooking SOA.

### 3.3    Contribution of aldehydes to observed oxidation products and total SOA
Since cooking oil vapors are comprised of a number of reactive aldehydes that can lead to SOA formation, we conduct
further experiments of SOA formation from these precursors and identify the relative contributions to observed
oxidation products and to total SOA. These results are applied to the heated cooking oil experiments to understand the
role of aldehydes in the overall production and evolution of cooking oil SOA.
#### 3.3.1    Formation of particle-phase oxidation products
As described in the earlier sections, we are able to quantify the mass concentrations of different compound groups (6
different combinations of functional groups, from C2 to C9, summarized in Table S2) in the particle phase for all
experiments. We denote the observed mass concentrations of compound group $i$ in SOA from canola oil
photooxidation as $M_i^{oil}$. The expected precursors to these oxidation products are likely aldehydes, since aldehydes are
emitted in significant amounts and are highly reactive. To examine this hypothesis, here we calculate the formation
of these observed compound groups from oxidation of aldehydes. For this calculation, heptanal, *trans*-2-heptenal,
*trans*-2-octenal, and *trans,trans*-2,4-heptadienal (Sigma Aldrich Co.) were considered because these aldehydes are the



dominant VOC precursors emitted from heated canola oil in our experiments as shown in Fig. S8. Unlike previous
work by Fullana et al. (2004b) and Klein et al. (2016a), gas-phase concentrations of decadienals were minimal in our
experiments. More volatile aldehydes, such as acrolein and methacrolein, were likely present but could not be captured
and analyzed by our techniques. The molar amount reacted for each aldehyde $j$ in the canola oil oxidation experiments
is denoted as $\Delta VOC_j^{oil}$, and was calculated based on the measured OH exposure.
In order to estimate the contribution from oxidation of an aldehyde $j$ in the gas-phase mix to the formation of each
compound group $i$, we conducted a series of experiments in which a representative aldehyde was oxidized, and the
molar yields of the various compounds were measured:
$\gamma_{ij} = \dfrac{M_{ij}^{ind}/MW_i}{\Delta VOC_j^{ind}}$ (3)
where $\gamma_{ij}$ represents the molar yield of compound group $i$ from precursor $j$, $M_{ij}^{ind}$ denotes the mass concentration of
compound $i$ observed in photooxidation experiments in which aldehyde $j$ was the sole precursor, $MW_i$ is the molecular
weight of compound $i$, and $\Delta VOC_j^{ind}$ is the amount of precursor $j$ reacted in each experiment. $\gamma_{ij}$ is then applied to the
heated cooking oil experiments to estimate the amount of oxidation products that would form from each precursor:
$M_i^{sum} = \sum_j \gamma_{ij} \Delta VOC_j^{oil} MW_i$ (4)
A sample calculation for this analysis is presented in Sect. 3 of SI. The comparison between $M_i^{sum}$ (contribution of
aldehyde oxidation to formation of compound $i$) and $M_i^{oil}$ (observed concentrations of compound $i$) is shown in Fig.
6. Based on this methodology, oxidation of aldehydes accounts for 56 µg m$^{-3}$ ($M_i^{sum}$) of the observed 75 µg m$^{-3}$ ($M_i^{oil}$)
(or 75%) particle-phase oxidation products measured at an OH exposure of $6.43\times10^{10}$ molecules cm$^{-3}$ s. The
contributions of alkanals (heptanal), alkenals (heptenal + octenal) and alkadienals (heptadienal) are 7%, ~31% and
37%, respectively.
While the amount of oxidation products expected from aldehydes is somewhat lower than that observed in canola oil
SOA, this difference may arise from differences in gas-particle partitioning between single aldehyde photooxidation
and canola oil photooxidation. As shown in Fig. S9, more oxygenated compounds (higher O:C and greater number of
functional groups) tend to be more abundant in the canola oil SOA than expected from aldehyde photooxidation,
suggesting that canola oil SOA is more favorable for oxygenated compounds to partition than SOA from individual
aldehydes. On the other hand, there is no clear trend in partitioning with respect to vapor pressures and carbon number.
It should be noted that uncertainties in the fitting procedure or estimation in the pseudo molecular ion (refer to Table
S2 and Fig. S5) can also result in uncertainties between -40% and +20%. Therefore, in summary, the quantified
oxidation products from canola oil SOA are generally consistent with those from aldehyde photooxidation, and the
relative amounts may be subject to further changes due to gas-particle partitioning.



### 3.3.2  Using the statistical oxidation model (SOM) framework

To further explore the evolution of canola oil SOA, we applied our results to the statistical oxidation model (SOM) framework developed by Cappa and Wilson (Cappa et al., 2013; Cappa and Wilson, 2012). SOM describes the oxidation chemistry of a VOC precursor through multi-generational space defined by the number of carbon and oxygen atoms present in the precursor and its possible SOA product molecules. The SOM does not specifically track the product composition in terms of functional groups, but provides adequate details to represent key atmospheric processes such as gas-particle partitioning, fragmentation, functionalization, reactions with oxidants, condensed-phase chemistry. The model has been applied to chamber experiments to derive parametrizations by fitting experimental data to both SOA mass concentration and the bulk aerosol O/C ratio. Eluri et al. (2018) used the chamber derived parameterizations to predict the properties of SOA generated from diesel exhaust in an oxidation flow tube reactor.

To the best of the authors' knowledge, there are no parameterizations for the oxidation of aldehydes. Therefore, in this study we first derived the parameterizations for aldehyde oxidation, and then use these parameters to predict the SOA mass concentrations. In order to obtain the parameters, we fit the measured SOA concentration from oxidation of heptanal, *trans*-2-heptenal, *trans,trans*-2,4-heptadienal at different OH exposures to optimize the six tunable parameters under low-NO$_x$ conditions (shown in Fig. S10). Best fit SOM parameters indicate that photooxidation leads to fragmentation per reaction with OH, as shown by a lower *mfrag* than compared to other systems e.g. alkanes ($\geq$2 for branched, cyclic or *n*-alkane under low-NO$_x$ conditions (Eluri et al., 2018)). Since a lower value for *mfrag* represents greater fragmentation (Cappa and Wilson, 2012), this again reflects the higher propensity for fragmentation in this SOA system. The optimized parameters were then used to predict the SOA concentration for canola oil photooxidation under different aging conditions.

Based on these established parameterizations for different aldehydes, model simulations were conducted for canola oil having a mixture of aldehydes under different photochemical aging conditions. It should be noted that we used parameterizations of heptanal for all alkanals, heptenal for all alkenals, and heptadienal for alkadienals. As shown in Fig. 7, the model generally captures the amount of SOA formed to within 50%, but overpredicts SOA formation at lower photochemical ages and underpredicts SOA concentrations at higher photochemical ages. In addition, SOM also tracks atomic O/C ratio which were further compared with the measured O/C ratio. SOM predicts an O:C around 0.7, which lies within ±20% of the measured O:C likely suggesting that the changes in chemical composition of cooking SOA is in good agreement with the model predictions.

One inconsistency between the model and measurements is the slope at which SOA is being formed. The experimental data suggest a steeper trend of SOA formation while the model predicts a more gradual increase in SOA formation. A potential explanation for this discrepancy is the contribution from other unmeasured VOCs. These VOCs are less reactive than those considered in the model, such that they contribute to higher SOA at higher OH exposures. Alternatively, these missing VOCs are more volatile such that more of their SOA is formed at later generations of oxidation. For example, acrolein forms SOA with measurable yields (Chan et al., 2010) and is emitted at large amounts from heated cooking oils (Klein et al., 2016a). Despite these limitations, these parameterizations generally capture the





amount of SOA formed and its degree of oxidation (O/C) on oxidation timescales relevant to urban areas (within 2
days) and are useful for representing cooking oil emissions in the chemical transport models. Overall, the amount of
SOA formed and the evolution upon oxidation can be well described by photooxidation of aldehydes.

**4      Conclusions and implications**
In this work, we characterized the detailed chemical composition of SOA generated from cooking oil vapors. We
showed that cooking SOA occurring as highly complex mixture can be deconvoluted using mass spectral
fragmentation pattern to extract useful information about the chemical identities of organic compounds, such as
functional groups and carbon number. Using this detailed chemical composition of cooking SOA, we showed that
fragmentation is an important pathway for oxidative processing of cooking emissions in the atmosphere even within
short timescales of oxidation. Furthermore, we showed that aldehydes can reasonably explain the formation of SOA
generated from cooking oil vapors and the oxidative evolution as described using a multi-generational oxidation
model. Our study, therefore, highlights the importance of molecular composition in constraining the chemical
properties of cooking SOA, as well as understanding the contribution of aldehydes in formation of SOA from cooking
emissions.
Consistent with other studies, our work has shown that aldehydes are an important class of VOC precursors emitted
from cooking emissions, and substantial efforts have been made to measure their emission factors depending on
different cooking settings (heating temperature, cooking style, food, ingredients) (Klein et al., 2016a, 2016b).
However, the contribution of aldehydes from cooking emissions is underrepresented in chemical transport models.
Recently, McDonald et al. (2018) showed that the ambient concentrations of OA were underpredicted when aldehydes
were not included in the box model calculations, suggesting that aldehydes, likely from food cooking, play an
important role in atmospheric oxidation chemistry. Furthermore, Klein et al. (2019) showed that heavy polluters like
restaurants play a significant role in contributing to the ambient cooking organic aerosol concentrations. In this study,
we showed a large fraction of the SOA is derived from aldehyde precursors, with strong similarities in chemical
composition. Therefore, it is important to consider the contribution of aldehyde chemistry in atmospheric models
towards total OA budget. Furthermore, we demonstrated the importance of fragmentation reactions and their influence
on OA properties such as volatility and chemical composition. Future work should therefore focus on measuring not
only the SOA formation, but also the oxygenated VOCs formed due to fragmentation upon aging to provide insights
into aging of cooking emissions.
Gas-particle partitioning of SOA can be further affected by non-ideal mixing, as well as morphology of the particles
(Shiraiwa et al., 2013; Zuend and Seinfeld, 2012). Future work should investigate the effect of these parameters on
cooking SOA properties and formation potential. To account for thermodynamic mixing favourability of the particles,
Hansen solubility framework developed by Ye et al. (2016) can be implemented to provide insights into SOA mixing
and yield enhancement. As shown in Ye et al. (2018) primary meat-cooking emissions can enhance SOA yield from
α-pinene due to similarity in Hansen solubility parameters suggesting that primary meat cooking particles are miscible



with α-pinene SOA. It should be noted that present study did not investigate the effect of atmospherically relevant
seed particles as well as NO$_x$ levels which are representative of typical urban environments. Since emissions upon
entering the atmosphere gets mixed with background air, other source emissions, and diluted upon mixing thereby
altering the gas-particle partitioning, and thus the total OA loading. Therefore, it is important to understand the changes
in partitioning and miscibility of cooking emissions as the composition continually evolves with atmospheric
processing. Additionally, as mentioned earlier cooking SOA undergoes large mass transfer limitations due to changes
in the phase state of the SOA particles, making it more so important to experimentally determine the corresponding
viscosity of cooking SOA. Therefore, future work should focus on measuring both the viscosity and miscibility of
SOA derived from cooking emissions.

*Data availability*. The data are available upon request to the corresponding author.

*Competing interests*. The authors declare that they have no conflict of interests.

*Acknowledgements*. The authors acknowledge Environment and Climate Change Canada (ECCC) for funding support
through the Government of Canada Grants and Contribution program. The authors would like to thank Shao-Meng Li
from ECCC for use of the thermodenuder, Chris Cappa from UC Davis for help with SOM simulations, Greg Evans,
Jeff Brook and Tengyu Liu from University of Toronto for helpful discussion.





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




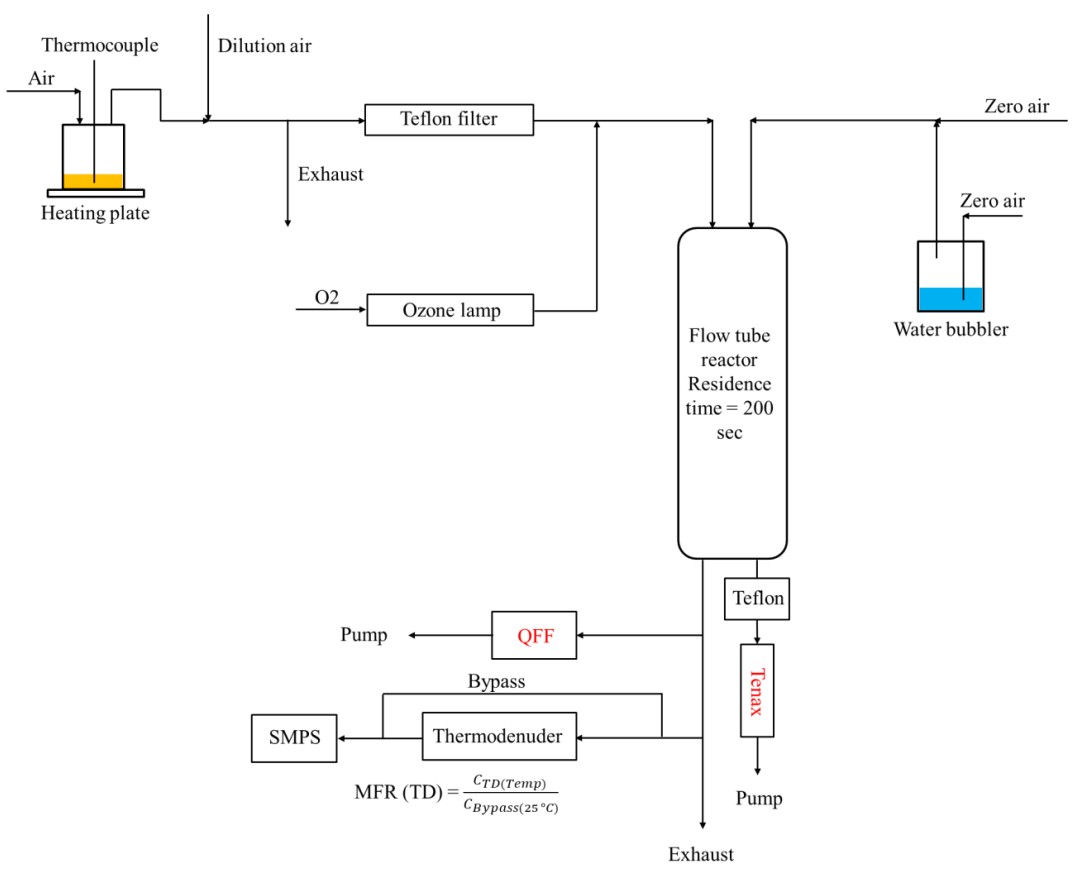


**Figure 1.** Experimental setup for oxidation of heated cooking oil emissions.



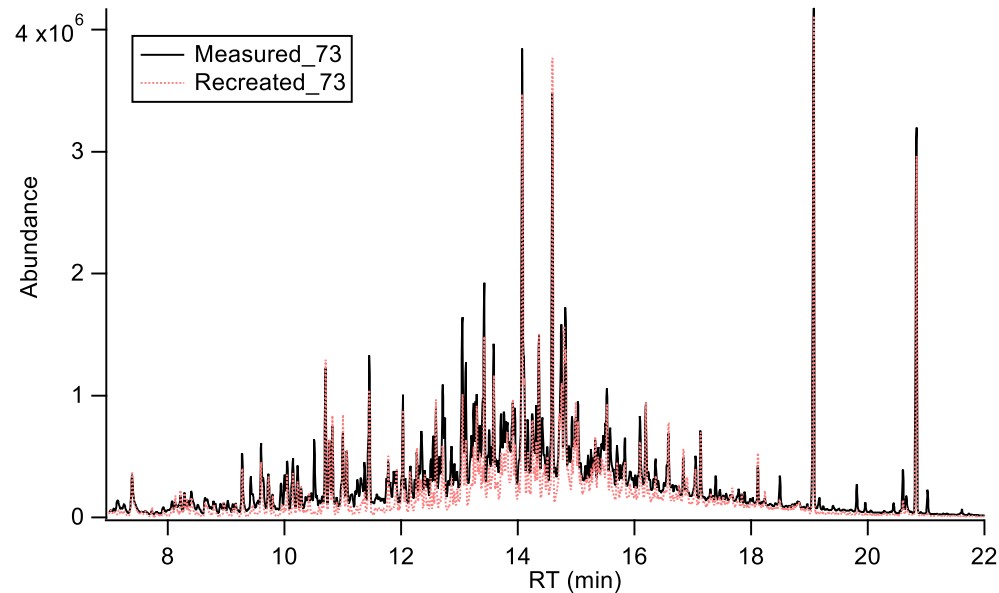


**Figure 2.** Highly complex mixture of canola oil SOA generated upon photooxidation. With known signal and mass fragmentation,
signal of *m/z* 73 can be recreated based on pseudo parent ions (e.g. M-15 used in this study).





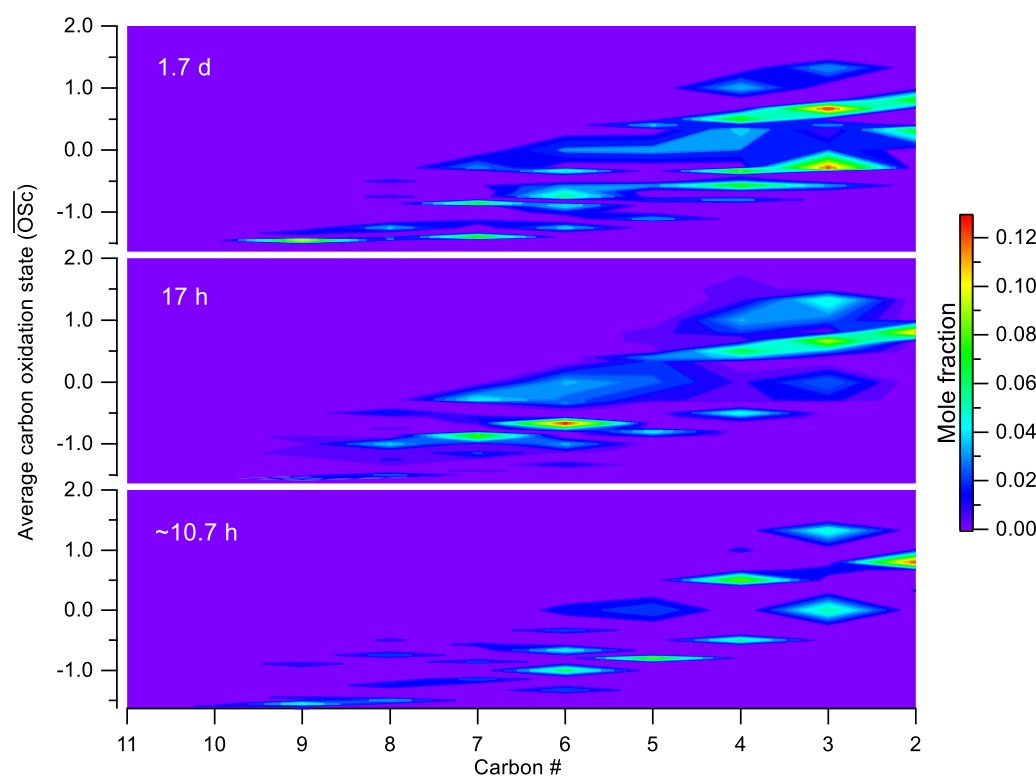


**Figure 3.** Evolution in $\overline{OSc}$-nc space for canola oil SOA under different conditions of photochemical aging. As the oxidation

progresses in the atmosphere, more compounds are formed with smaller nc and higher $\overline{OSc}$ suggesting fragmentation to be a

dominant pathway of oxidation for cooking emissions in the atmosphere.


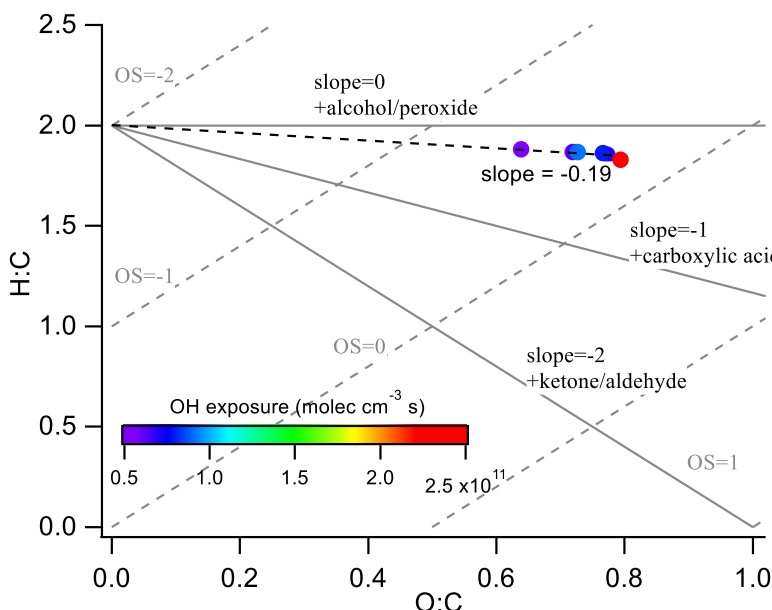


**Figure 4.** Van Krevelen diagram of canola oil SOA coloured by different OH exposure. In the background, average carbon

oxidation state ($\overline{OSc}$) and functionalization slopes are shown for reference. The slope of -0.19 for canola oil SOA corresponds to

formation of both alcohol and carboxylic acid consistent with the chemical composition obtained from TD-GC/MS.




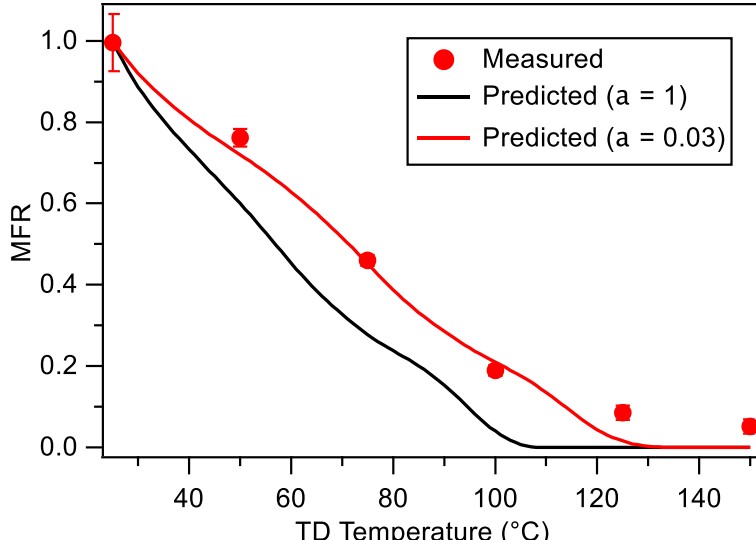


**Figure 5.** Mass thermogram of canola oil SOA at an OH exposure of $9.23 \times 10^{10}$ molecules cm$^{-3}$ s. The black line represents model simulations using $\alpha = 1$ underpredicting the measured MFR. The red line corresponds to model simulations using $\alpha = 0.03$ predicting the measurements reasonably well, therefore implying kinetic limitations in the system. The error bars represent $\pm 1\sigma$.






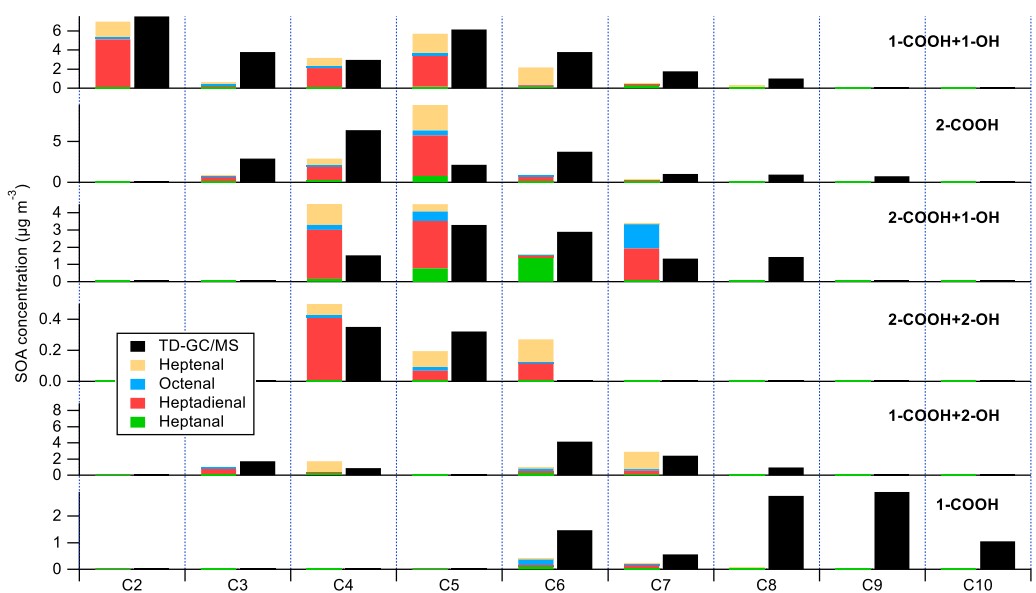


**Figure 6.** Prediction of different compounds formed at an OH exposure of $6.43 \times 10^{10}$ molecules cm$^{-3}$ s using product molar yields
of heptanal, heptenal, octenal, and heptadienal. The total aldehydes products can explain the observed oil SOA products within a
factor of half, while the inconsistency in prediction of some SOA products is likely caused by differences in gas-particle partitioning
in both photooxidation systems.

604

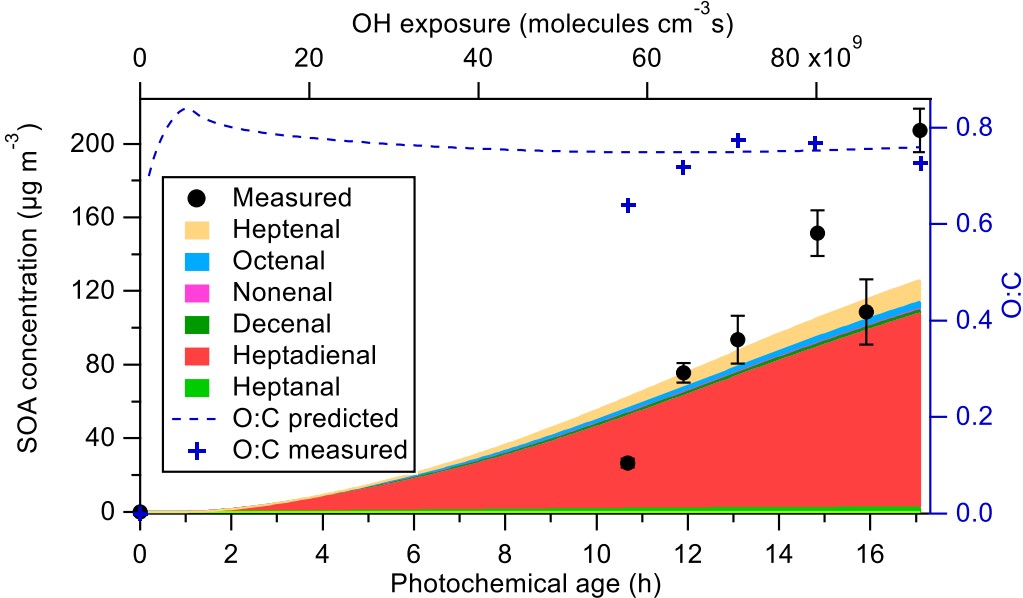

**Figure 7.** SOM prediction of SOA produced from different aldehydes with increasing photochemical age. The model overpredicts SOA formation at lower photochemical age, while underpredicts SOA formation by ~40% at higher photochemical age, suggesting that traditional VOC precursors cannot fully explain the SOA formation, and other gas-phase precursors maybe needed to better constrain the formation of SOA at higher aging conditions. In addition, the SOM predicted O:C is within ±20% of the measured O:C suggesting that the overall change in chemical composition of cooking SOA is predicted reasonably well.