# Peer review of "Characterization of secondary organic aerosol from heatedcooking oil emissions: evolution in composition and volatility"

_Atmospheric Chemistry and Physics, 2020_

## Referee Comment (RC2) · Anonymous Referee #2 · 9 Nov 2020

The manuscript by Takhar et al. reported the SOA formation from OH oxidation of heated cooking oil and characterized the SOA composition and volatility using TD-GC/MS. A new method was used to analyze the composition of the complex product mixtures. The authors have found that an increase in OSc and a decrease in carbon number upon oxidation, which was attributed to fragmentation reactions during OH aging. In addition, by comparing product yields from individual precursors and applying an oxidation model, they conclude that aldehyde precursors are the main contributors of SOA formed from heated cooking oil. Overall, this is a well-written manuscript and the results can be a useful addition to better understanding the formation, composition and volatility of cooking SOA. I would recommend publication after a minor revision.

**Specific Comments**

1. The authors stated the increase in OSc and decrease in nC with oxidation. However, it is not very clear from Figure 3. Could the authors add the average OSc and nC in every panel of Figure 3?

2. L193-197, the authors did not compare the same thing here. In Lambe et al. (2012, 2015), they found the evidence of fragmentation because of the decrease in SOA yield upon oxidation. However, in this study, the evidence of fragmentation is from the decrease in average carbon number. If the authors also look at the SOA yield, it never decrease with increasing photochemical age (Figure 7 and Table S1). Therefore, the authors could not state that fragmentation reactions happen earlier for cooking oil than other precursors (e.g., alkanes and isoprene).

3. L293-295, I do not see any O/C or number of functional group dependence in Figure S9. The authors might need to change their way to present these data.

4. Figure 6 looks very interesting, but the authors did not discuss much about it. It seems that for higher-carbon-number products and 1-COOH products, the agreement is worse than others compounds. Any explanation for that?

5. In Figure 2, the recreated m/z 73 seems agree well with the measured signal. Could the authors provide a scatter plot as well? Maybe it can replace Figure S4c.

6. L201, what are the O/C ratios in previous studies? The author should put them here for comparison.

7. Exp. 7 (photochemical age = 40.7 h) is not included in Figure 7. The authors should mention that somewhere.

**Technical Corrections**

L470-475, same references.

There is an incorrect number (1:2:1) in Figure S5.

---

## Author Comment (AC1) · 7 Feb 2021

The authors thank the reviewers for their insightful comments. The response to each comment is in blue with corresponding changes in the manuscript italicized.

**Anonymous Referee #1**

The manuscript presents some interesting measurements of cooking SOA by TDGC/MS. The authors adopt a new method to achieve molecular speciation of the SOA, where SOA complex mixture was deconvoluted using mass spectral fragmentation patterns to extract useful information about functional groups and carbon numbers. They also derived the parameterizations for aldehyde oxidation and used the derived parameters to predict SOA mass. The model results generally well captured the amount of SOA formed and its chemical characteristics, e.g., O/C. Overall, this paper makes a valuable contribution to current cooking SOA understanding.

Q1. Hydroxyl radicals as major oxidants are produced through the ozone photolysis in the flow tube. However, ozone concentration would be high in the flow tube. What is the influence of ozone oxidation of aldehydes in the experiments?

We agree that ozone concentration would be high in the flow tube. However, we did not observe any particle formation during ozonolysis of heated cooking oil where OH radicals were not present. Since aldehydes are major contributors of gas-phase emissions from cooking oils, we believe that during photooxidation of aldehyde precursors ozone chemistry would have negligible effect on particle formation. Furthermore, gas-phase reaction rate constant of aldehydes with ozone is of the order of $10^{-17}$-$10^{-18}$ (Atkinson and Arey, 2003; Atkinson and Carter, 1984) much lower than that of hydroxyl radicals, so the timescale for ozonolysis of aldehydes would likely be longer than the residence time in the flow tube reactor.

To understand the reactivity of C=C in unsaturated aldehydes, methacrolein is used as an example calculation to provide insights into the reaction timescales as its reaction rate constant with both OH and ozone is readily available. Following calculations shows that timescale of reaction with ozone is much slower than the residence time in the flow tube suggesting that the ozonolysis of unsaturated aldehydes play a negligible role in the formation of SOA from aldehydes or cooking emissions.

The reaction rate constant of methacrolein is obtained from Atkinson and Arey, 2003.

$k_{OH}$ is 2.9E-11 $cm^3$ $molec^{-1}$ $s^{-1}$

$k_{O_3}$ is 1.2E-18 $cm^3$ $molec^{-1}$ $s^{-1}$.

For the lowest OH exposure, OH conc = 2.88E8 molec $cm^{-3}$ and ozone conc is 0.5 ppm. At these conditions, methacrolein reaction timescale with OH is $1/k_{OH}*[OH]$ = ~120 s, while reaction timescale with ozone is 1129 min.

For the highest OH exposure, OH conc = 1.1E9 molec $cm^{-3}$ and ozone conc is 12.6 ppm. At these conditions, methacrolein reaction timescale with OH is 31 s, while reaction timescale with ozone is ~45 min.

To clarify this, following text have been added to the manuscript in L91-94, as well as above sample calculation have been added to SI in Sect. 1 in L60-73:

*"…The effect of ozone on the SOA formation was found to be negligible as the reaction timescales of aldehydes with ozone were calculated to be at least 100 times longer than those with OH. A*

*sample calculation for methacrolein reaction timescales with OH and ozone is shown in SI in Sect. 1."*

Q2. What are the criteria in selecting the model compounds (Table S2) in each functional group class (e.g., different carbon number range used for different classes)?

Based on the carbon number range of VOC precursors emitted from cooking emissions, the resulting SOA products will have a similar or lower carbon # assuming oligomerization reactions are not dominant in the system. We also expect that photooxidation reactions will lead to addition of -OH and =O groups based on knowledge about gas-phase oxidation chemistry. Based on these different combinations of functional groups we selected the model compounds in Table S2 to characterize the composition of cooking SOA.

Q3. Figure 3: The author mentioned there is an increase in the average oxidation state (from -0.6 to -0.24) and a decrease in the average carbon number (from 5.2 to 4.9) with increasing photochemical aging (line 15 and 1ine 188-193 and Fig.3 caption). The decrease in carbon is not so significant, and from the figure, the mole fraction of carbon 7-9 compounds is even higher in the 1.7d photochemical aging condition than those in 10.7h condition. The conclusion of dominant fragmentation should be better elaborated. The SOA concentration actually increased with further oxidation (TableS1).

We agree that the decrease in average carbon # is not significant in this study. However, the effect of fragmentation is evident from an increasing fraction of smaller and more oxygenated compounds formed during photooxidation. For instance, the total fraction of C2-C7 SOA products increased from 81% in 10.7h SOA to 89% in 1.7d SOA. Of this fraction, the smaller carbon # compounds (C2-C4) which are indicative of fragmentation processes increased from 42% in 10.7h SOA to ~49% in 1.7d SOA. On the other hand, total fraction of >C7 (C8-C10) products declined from ~19% to ~11% as SOA aged. An increase in smaller and more oxygenated compounds, along with decrease in larger and less oxygenated products from 10.7h SOA to 1.7d SOA suggests that fragmentation reactions are responsible for the shift towards formation of smaller oxygenated compounds. We also use the term fragmentation to refer strictly to the decrease in carbon number, and not decrease in SOA concentration. Therefore, increase in SOA concentration is not strictly inconsistent with fragmentation, since the effect of increased $\overline{OSc}$ on volatility can exceed that of decreased carbon number. To clarify this, the following edits have been made to the manuscript in L193-198:

*"...comprised of*  *~19% larger (C8–C10) and less oxygenated compounds, this fraction declined to ~11% at higher OH exposures. Furthermore, the total fraction of C2-C7 products increased from 81% to 89% when OH exposure increased from 10.7 h to 1.7 d. Of this fraction, the smaller carbon # compounds (C2-C4) which are indicative of fragmentation processes increased from 42% at 10.7 h to ~49% at 1.7 d. An increase in smaller and more oxygenated compounds, along with decrease in larger and less oxygenated products suggests that fragmentation reactions are responsible for the shift towards formation of smaller oxygenated compounds."*

Q4. Figure 5 compares the measured and modeled mass thermograms for canola oil SOA. A mass accommodation coefficient of 0.03 was used in the model prediction. How sensitive is the predicted results to the accommodation coefficient? When using different accommodation coefficients of 1 and 0.03, the differences between MFR are only around 30% (at the same TD temperature). Maybe be better to add some thermograms using middle accommodation coefficient values between 1 and 0.03.

Thanks for the suggestion. We have modified Fig. 5 in the manuscript with different accommodation coefficients in the range 1-0.03.

[Figure]

Q5. Line 291-294: Other unidentified I/SVOCs may also play a role in the unexplained SOA mass.

Thanks for the suggestion. It is likely that other unidentified precursors such as IVOCs or SVOCs can contribute to the unexplained SOA. However, IVOCs from cooking emissions have not yet been positively identified. Based on the reviewer suggestion, the following changes have been made to the manuscript in L342-347:

*"...model predictions. Furthermore, the unexplained SOA can likely arise from other unidentified S/IVOCs as hypothesized by Liu et al. (2017c). However, unlike traffic emissions (Zhao et al., 2014), S/IVOCs from cooking has not been positively identified. In addition, small VOC precursors like acrolein and malondialdehyde which have been measured in large quantities from cooking emissions (Klein et al., 2016a), may form SOA products having higher O/C ratios, which may better explain the O/C ratios observed in our experiments."*

In addition, please also refer to response to Q7, Reviewer #2.

Q6. Line 314-321: When deriving the parameters for aldehyde oxidation, six tunable parameters were used to fit the measured SOA concentration. Compared with other systems, a lower mfrag was used in this study. The author attributed this to the greater fragmentation in this SOA system. However, the relative strength of these six parameters used to fit SOA concentrations is different for heptanal, 2-heptenal and 2,4- heptadienal oxidation experiments (Fig. S10). For example, mfrag used in heptanal experiment data is one order of magnitude lower than that used for the other two aldehydes. In addition to mfrag, what are the corresponding processes of the other five parameters? Are those related to gas-particle partitioning, functionalization, reactions with oxidants, or condensed-phase chemistry mentioned earlier (line 308-309)? It would be interesting to discuss the relationship between these parameters and their corresponding chemical or physical processes and how they behave in this system.

The six tunable parameters in SOM are: mfrag- which describes fragmentation reactions, ΔLVP- is decrease in logarithm of volatility upon addition of oxygen atom. Pox1, Pox2, Pox3, Pox4- describes addition of 1, 2, 3, and 4 oxygen atoms per reaction with OH, respectively.
The probability of a reaction with OH leading to fragmentation is calculated as $P_{frag} = (O:C)^{mfrag}$, where mfrag is the fitting parameter. The corresponding functionalization probability can be estimated as $P_{func} = 1 - P_{frag}$. Instantaneous gas/particle partitioning equilibrium is inherently assumed at every timestep in the model. Addition of oxygen atoms upon reaction with OH will govern the reaction with oxidants. Condensed-phase chemistry was not considered in the model.
To answer reviewer's question, as the # of oxygen atom addition increases on carbon backbone, it will likely result in the higher fragmentation probability. Therefore, as shown in Fig. S10, Pox4 is much higher for heptanal than 2-heptenal or 2,4-heptadienal, thereby having much lower mfrag value than 2-heptenal or 2,4-heptadienal.

Q7. In real cooking emissions, POA are also emitted with the aldehydes and other gasphase precursors. It would be useful if the authors can project how the inclusion of POA in the system would affect their results of O/C, etc.

Inclusion of POA during oxidation of cooking vapors will likely decrease the overall O:C (or $\overline{OSc}$) of the system as POA is less functionalized than SOA due to higher contributions from long chain fatty acids, such as C16, C18 thereby giving rise to $\overline{OSc}$ of POA from heated cooking oils ~ -1.7 (Takhar et al., 2019) which is much lower compared to $\overline{OSc}$ of cooking SOA measured in this study. Therefore, inclusion of POA would likely lead to an overall decrease in the average $\overline{OSc}$ or O:C of the system. However, it should be noted that POA can itself undergo heterogenous oxidation reactions in the atmosphere resulting in an increase in O:C. On the other hand, other gas phase precursors that can potentially contribute to total vapor emissions from cooking could be S/IVOCs which have not been positively identified from cooking emissions, but have been shown to contribute to SOA from other sources e.g. traffic (Zhao et al., 2014). A similar projection for cooking emissions can likely be made. Furthermore, depending on the cooking conditions, it has been shown that cooking can emit large amounts of terpenes upon addition of condiments or spices to heated cooking oils. Emissions of terpenes have been shown to significantly contribute to total SOA production (Klein et al., 2016b; Liu et al., 2017a). Based on the reviewer suggestion, the following edits have been made to the manuscript in L384-391:

*"…Formation of SOA from cooking emissions in the atmosphere is likely influenced by emissions of POA, and other gas-phase precursors. Therefore, inclusion of POA during atmospheric processing of cooking emissions will likely influence the physicochemical properties of cooking SOA. For instance, with cooking POA being much less functionalized than SOA, inclusion of POA will likely decrease the system O:C (or $\overline{OSc}$). However, POA from cooking emissions can undergo heterogeneous reactions in the atmosphere, thereby increasing O:C (or $\overline{OSc}$). On the other hand, there could potentially be contributions from other gas-phase precursors or S/IVOCs emitted from cooking vapors that can result in SOA formation. These precursors can potentially contribute to SOA formation from cooking emissions, but their oxidative evolution in the atmosphere is not well understood."*

References

Atkinson, R. and Arey, J.: Atmospheric Degradation of Volatile Organic Compounds, Chem. Rev., 103(12), 4605–4638, doi:10.1021/cr0206420, 2003.

Atkinson, R. and Carter, W. P. L.: Kinetics and Mechanisms of the Gas-Phase Reactions of Ozone with Organic Compounds under Atmospheric Conditions, Chem. Rev., 84(5), 437–470, doi:10.1021/cr00063a002, 1984.

Klein, F., Platt, S. M., Farren, N. J., Detournay, A., Bruns, E. A., Bozzetti, C., Daellenbach, K. R., Kilic, D., Kumar, N. K., Pieber, S. M., Slowik, J. G., Temime-roussel, B., Marchand, N., Hamilton, J. F., Baltensperger, U., Prévôt, A. S. H. and El Haddad, I.: Characterization of Gas-Phase Organics Using Proton Transfer Reaction Time-of-Flight Mass Spectrometry: Cooking Emissions, Environ. Sci. Technol., 50(3), 1243–1250, doi:10.1021/acs.est.5b04618, 2016a.

Klein, F., Farren, N. J., Bozzetti, C., Daellenbach, K. R., Kilic, D., Kumar, N. K., Pieber, S. M., Slowik, J. G., Tuthill, R. N., Hamilton, J. F., Baltensperger, U., Prévôt, A. S. H. and El Haddad, I.: Indoor terpene emissions from cooking with herbs and pepper and their secondary organic aerosol production potential, Sci. Rep., 6, 1–7, doi:10.1038/srep36623, 2016b.

Liu, T., Liu, Q., Li, Z., Huo, L., Chan, M. N., Li, X., Zhou, Z. and Chan, C. K.: Emission of volatile organic compounds and production of secondary organic aerosol from stir-frying spices, Sci. Total Environ., 599–600, 1614–1621, doi:10.1016/j.scitotenv.2017.05.147, 2017a.

Liu, T., Wang, Z., Huang, D. D., Wang, X. and Chan, C. K.: Significant Production of Secondary Organic Aerosol from Emissions of Heated Cooking Oils, Environ. Sci. Technol. Lett., 5(1), 32–37, doi:10.1021/acs.estlett.7b00530, 2017c.

Takhar, M., Stroud, C. A. and Chan, A. W. H.: Volatility Distribution and Evaporation Rates of Organic Aerosol from Cooking Oils and their Evolution upon Heterogeneous Oxidation, ACS Earth Sp. Chem., 3(9), 1717–1728, doi:10.1021/acsearthspacechem.9b00110, 2019.

Zhao, Y., Hennigan, C. J., May, A. A., Tkacik, D. S., De Gouw, J. A., Gilman, J. B., Kuster, W. C., Borbon, A. and Robinson, A. L.: Intermediate-volatility organic compounds: A large source of secondary organic aerosol, Environ. Sci. Technol., 48(23), 13743–13750, doi:10.1021/es5035188, 2014.

**Anonymous Referee #2**

The manuscript by Takhar et al. reported the SOA formation from OH oxidation of heated cooking oil and characterized the SOA composition and volatility using TD-GC/MS. A new method was used to analyze the composition of the complex product mixtures. The authors have found that an increase in OSc and a decrease in carbon number upon oxidation, which was attributed to fragmentation reactions during OH aging. In addition, by comparing product yields from individual precursors and applying an oxidation model, they conclude that aldehyde precursors are the main contributors of SOA formed from heated cooking oil. Overall, this is a well-written manuscript and the results can be a useful addition to better understanding the formation, composition and volatility of cooking SOA. I would recommend publication after a minor revision.

**Specific Comments**

1. The authors stated the increase in OSc and decrease in nC with oxidation. However, it is not very clear from Figure 3. Could the authors add the average OSc and nC in every panel of Figure 3?

Please refer to response to Q3, Reviewer #1.

2. L193-197, the authors did not compare the same thing here. In Lambe et al. (2012, 2015), they found the evidence of fragmentation because of the decrease in SOA yield upon oxidation. However, in this study, the evidence of fragmentation is from the decrease in average carbon number. If the authors also look at the SOA yield, it never decrease with increasing photochemical age (Figure 7 and Table S1). Therefore, the authors could not state that fragmentation reactions happen earlier for cooking oil than other precursors (e.g., alkanes and isoprene).

We agree that fragmentation reactions in this study are because of decrease in carbon # and not due to decrease in SOA yield. As suggested by the reviewer, L193-197 (now updated as L200-204) have been removed from the manuscript.

3. L293-295, I do not see any O/C or number of functional group dependence in Figure S9. The authors might need to change their way to present these data.

Figure S9(b) and (c) have been replaced with following figures, where y-axis is replotted as the ratio of aldehydes products to canola oil products instead of aldehydes products only. Since most of the products ratio lies below y-axis = 1 line suggesting that more oxygenated products partitions readily in canola oil SOA than individual aldehydes SOA. Please note that L293-295 is updated as L306-308.

Furthermore, Fig. S9(a) and (d) have also been updated with the similar y-axis against vapor pressure and carbon #.

[Figure]

4. Figure 6 looks very interesting, but the authors did not discuss much about it. It seems that for higher-carbon-number products and 1-COOH products, the agreement is worse than others compounds. Any explanation for that?

The formation of higher carbon number products from the precursors photo-oxidized in this study were not observed likely due to negligible oligomerization reactions or reactions occurring in the particle-phase that can form SOA products with higher carbon # than the smaller carbon # parent VOCs photo-oxidized in this study. In addition, 1-COOH compounds are likely to be present as

primary vapors in the gas-phase which can subsequently partition to the condensed phase upon SOA formation. Based on the reviewer comment, the following edits have been made to the manuscript in L302-305:

*"...canola oil photooxidation. As shown in Fig. 6, the formation of higher carbon # products cannot be explained from the photooxidation of aldehydes used to predict oil oxidation products likely due to the assumption of negligible particle-phase or oligomerization reactions occurring in the condensed phase. In addition, higher carbon # acids are likely present as primary vapors in the gas phase which can then partition to the condensed phase upon SOA formation."*

5. In Figure 2, the recreated m/z 73 seems agree well with the measured signal. Could the authors provide a scatter plot as well? Maybe it can replace Figure S4c.

Thanks for the suggestion. Fig. S4(c) is replaced with a scatter plot in the SI as shown below. However, it should be noted that a better assessment of recreated signal should be done by plotting chromatograms to evaluate any under/overestimation of the peaks or model compounds.

[Figure]

6. L201, what are the O/C ratios in previous studies? The author should put them here for comparison.

The O/C ratios reported in previous studies ranged between 0.24 to 0.46. The following edits have been made to the manuscript in L207-209:

*"...The O:C ratios measured using an AMS (Kaltsonoudis et al., 2017; Liu et al., 2017b) ranged between 0.24-0.46 which are within a factor of 2 measured in this study."*

7. Exp. 7 (photochemical age = 40.7 h) is not included in Figure 7. The authors should mention that somewhere.

Thanks for pointing this out. The model simulations were run with OH exposure in the range similar to that of aldehyde photooxidation (Fig. S10). Upon further examination, we discovered that the OH exposure in the model results were incorrectly plotted in Fig. 7. In addition, we found that in earlier experiments we were unable to measure decadienal, but upon further examination, we found decadienal can be captured on Tenax tubes by measuring for longer duration, and so we have now included decadienal as one of the SOA precursors and estimated the formation of speciated products and total SOA using SOM. We have made corresponding changes to Figs. 6, 7 and S8. After correcting this error, the following corrections have been made to the manuscript with Fig. 7 revised as shown below. Our updated SOM estimate of heptadienal SOA contribution is now ~19% (down from 35% in our previous estimate, which corresponded to SOA at higher OH exposure). At the same time, decadienal accounts for 38% of the SOA. The total estimated contribution from aldehydes to canola oil SOA is 62%. Furthermore, it should be noted the modeled O:C estimate declined from 0.7 to 0.51 with the inclusion of decadienal.

*In L334, "…aging conditions in the OH exposure range similar to that of aldehyde photooxidation."*

*In L338, "…Fig. 7, the model generally captures the amount of SOA formed to up to 62%, but…"*

*In L341-347 "…SOM predicts an O:C around 0.51, which  is within ±50% of the measured O:C likely suggesting that the changes in chemical composition of cooking SOA is in  a reasonable agreement with the model predictions. Furthermore, the unexplained SOA can likely arise from other unidentified S/IVOCs as hypothesized by Liu et al. (2017c). However, unlike traffic emissions (Zhao et al., 2014), S/IVOCs from cooking has not been positively identified. In addition, small VOC precursors like acrolein and malondialdehyde which have been measured in large quantities from cooking emissions (Klein et al., 2016a), may form SOA products having higher O/C ratios, which may better explain the O/C ratios observed in our experiments."*

[Figure]

**Technical Corrections**

L470-475, same references.

Thanks for pointing this out. Reference in line 473-475 (updated as L499-501) have been removed.

There is an incorrect number (1:2:1) in Figure S5.

Thanks for pointing this out. 1:2:1 has been corrected to 1.2:1.

References

Kaltsonoudis, C., Kostenidou, E., Louvaris, E., Psichoudaki, M., Tsiligiannis, E., Florou, K., Liangou, A. and Pandis, S. N.: Characterization of fresh and aged organic aerosol emissions from meat charbroiling, Atmos. Chem. Phys., 17(11), 7143–7155, doi:10.5194/acp-17-7143-2017, 2017.

Klein, F., Platt, S. M., Farren, N. J., Detournay, A., Bruns, E. A., Bozzetti, C., Daellenbach, K. R., Kilic, D., Kumar, N. K., Pieber, S. M., Slowik, J. G., Temime-roussel, B., Marchand, N., Hamilton, J. F., Baltensperger, U., Prévôt, A. S. H. and El Haddad, I.: Characterization of Gas-Phase Organics Using Proton Transfer Reaction Time-of-Flight Mass Spectrometry: Cooking Emissions, Environ. Sci. Technol., 50(3), 1243–1250, doi:10.1021/acs.est.5b04618, 2016a.

Liu, T., Li, Z., Chan, M. and Chan, C. K.: Formation of secondary organic aerosols from gas-phase emissions of heated cooking oils, Atmos. Chem. Phys., 17(12), 7333–7344, doi:10.5194/acp-17-7333-2017, 2017b.

Liu, T., Wang, Z., Huang, D. D., Wang, X. and Chan, C. K.: Significant Production of Secondary Organic Aerosol from Emissions of Heated Cooking Oils, Environ. Sci. Technol. Lett., 5(1), 32–37, doi:10.1021/acs.estlett.7b00530, 2017c.

Zhao, Y., Hennigan, C. J., May, A. A., Tkacik, D. S., De Gouw, J. A., Gilman, J. B., Kuster, W. C., Borbon, A. and Robinson, A. L.: Intermediate-volatility organic compounds: A large source of secondary organic aerosol, Environ. Sci. Technol., 48(23), 13743–13750, doi:10.1021/es5035188, 2014.